# Extracting average properties of disordered spin chains with translationally invariant tensor networks

Kevin Vervoort,[1] Wei Tang,[1] and Nick Bultinck[1]

[1]*Department of Physics and Astronomy, Ghent University, Krijgslaan 281, 9000 Gent, Belgium*
(Dated: May 1, 2025)

We develop a tensor network-based method for calculating disorder-averaged expectation values in random spin chains without having to explicitly sample over disorder configurations. The algorithm exploits statistical translation invariance and works directly in the thermodynamic limit. We benchmark our method on the infinite-randomness critical point of the random transverse field Ising model.

*Introduction* – Since the invention of the Density Matrix Renormalization Group (DMRG) algorithm in 1992 [1], a broad array of powerful tensor network-based methods has been developed for studying quantum many-body systems (see e.g. [2–6] and references therein). The community has reached a point at which low-temperature equilibrium properties of most local spin and fermion Hamiltonians on a one-dimensional lattice can be studied with rather limited computational resources, including many gapless systems. One notable exception, however, are systems with quenched randomness. The challenge in simulating these systems is two-fold. First, there is no translation symmetry such that the left-right sweeping procedure of the original DMRG algorithm appears to be the most appropriate way to optimize the Matrix Product State (MPS). But randomness can cause entanglement to be inhomogeneously distributed on different length scales (such as for example in random singlet states [7–9], which significantly increases the number of sweeps needed for convergence. Secondly, calculating disorder-averaged properties requires a large number of simulations with different disorder samples. Some interesting recent progress on the first challenge has been made via a numerical implementation of the rigorous renormalization group [10, 11], but the second challenge in dealing with a large number of disorder samples remains.

In this work we address this issue for low (but nonzero) temperature properties of random spin chains. We consider systems with statistical translation invariance, meaning that the distribution of the random variables is the same on every site, and represent the density matrix from which disorder-averaged expectation values are obtained as a Matrix Product Operator (MPO) [12]. MPOs are a straightforward generalization of MPS to operators, and by exploiting (statistical) translation invariance they can also be used directly in the thermodynamic limit while still permitting an efficient evaluation of expectation values. We present an algorithm to explicitly construct the desired MPO starting from the infinite-temperature state, and illustrate it on the random transverse field Ising model near criticality. Down to low temperatures we can reproduce the average properties of this paradigmatic model with relatively small

bond dimension.

*Algorithm* – We consider disordered Hamiltonians of the form $\mathcal{H}[\{R_n\}] = \sum_n \tilde{h}_n[R_n]$, where the integer $n$ labels the lattice sites and the local terms $\tilde{h}_n[R_n]$ act non-trivially on a finite number of consecutive spins starting at $n$. The local Hamiltonian terms depend on the quenched disorder variables $R_n$. In what follows we consider systems with discrete disorder where the disorder parameters $R_n$ can take on $N_D$ different values, with probability $P(R_n)$. We take $P(R_n)$ to be the same at every site, leading to statistical translation invariance. The main idea behind our algorithm is to approximate the mixture of Gibbs states with different quenched disorder configurations by a single MPO, which is translationally invariant due to the statistical translation invariance. Concretely, we devise an algorithm to approximate following density matrix:

$$\rho = \sum_{\{R_n\}} \left[ \prod_n P(R_n) \right] \rho_G[\{R_n\}], \qquad (1)$$

where the sum is over disorder configurations, and $\rho_G[\{R_n\}] = e^{-H[\{R_n\}]/T}/Z[\{R_n\}]$ with $T$ the temperature and $Z[\{R_n\}] = \text{tr}\left(e^{-H[\{R_n\}]/T}\right)$. It follows from the definition of $\rho$ that $\text{tr}(\rho O)$ is the disorder-averaged thermal expectation value of $O$. Before going into the details of the algorithm, let us note that (in contrast to clean spin chains [13, 14]) there is no a priori theoretical reason to assume that $\rho$ has an efficient MPO approximation. One of the main results of this work is therefore to provide numerical evidence that such an efficient approximation does indeed exist, and can capture the non-trivial physics of disordered spin chains.

As in Ref. [15] we introduce ancilla qudits $|R_n\rangle$ of dimension $N_D$ on every site, and define a translationally invariant Hamiltonian $H = \sum_n h_n$, where the local terms $h_n$ are

$$h_n = \sum_{R_n=1}^{N_D} \tilde{h}_n[R_n] \otimes |R_n\rangle\langle R_n| \qquad (2)$$

These terms represent a controlled action of the original terms $\tilde{h}_n[R_n]$ in the disordered Hamiltonian, where the

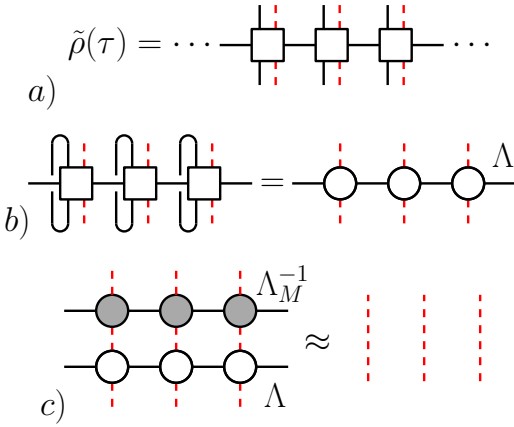

FIG. 1. (a) MPO representation of $\tilde{\rho}(\tau)$. Horizontal lines represent the virtual bonds. Vertical black full lines represent the physical spin indices. Red dashed lines act on the disorder qudits. (b) The MPO $\Lambda$ is obtained by tracing out the physical spin indices in $\tilde{\rho}(\tau)$. (c) An ansatz MPO $\Lambda_M^{-1}$ (filled circles) is used to approximately invert $\Lambda$.

ancilla qudits play the role of the control qudits. As with controlled quantum gates, the Hamiltonian terms $h_n$ are diagonal in the control (or disorder) qudits, and simply measure their value. Depending on this value a particular local Hamiltonian term of the disordered Hamiltonian acts on the physical spins.

It is important to note that $\rho$ as defined in Eq. (1) is *not* simply the Gibbs state of $H$ with traced out disorder qudits. This is because first constructing the Gibbs state of $H$ and then tracing out the disorder qudits fails to include the all-important normalization factors $1/Z[\{R_n\}]$. The algorithm we develop in this work is therefore a modified version of Time-Evolving Block Decimation (TEBD) [16], which constructs the Gibbs state of $H$ via imaginary-time evolution while at the same time also ensuring the correct normalization. During the imaginary-time evolution, we work with following operator:

$$\tilde{\rho}(\tau) = N(\tau)e^{-\tau H}, \qquad (3)$$

where $N(\tau) = N_R(\tau) \otimes \mathbb{1}_\sigma$ is a diagonal operator acting as the identity on the physical spins, and whose diagonal elements are given by $1/Z_\tau[\{R_n\}]$, i.e. the inverse partition functions at imaginary time $\tau$ for the different disorder configurations. The operator $N(\tau)$ thus ensures that tracing out the physical spins $\sigma_n$ in $\tilde{\rho}(\tau)$ produces an identity matrix on the disorder qudits: $\mathrm{tr}_{\{\sigma\}}\tilde{\rho}(\tau) = \mathbb{1}_R$.

At every step $\tilde{\rho}(\tau)$ will be represented as an infinite MPO [see Fig. 1 (a)]:

$$\tilde{\rho}(\tau) = \sum_{\{\sigma\},\{\sigma'\}} \sum_{\{R\}} \mathrm{tr}\left(\cdots A_{R_n}^{\sigma_n,\sigma'_n} A_{R_{n+1}}^{\sigma_{n+1},\sigma'_{n+1}} \cdots\right) \qquad (4)$$
$$\cdots |\sigma_n, R_n\rangle\langle\sigma'_n, R_n| \otimes |\sigma_{n+1}, R_{n+1}\rangle\langle\sigma'_{n+1}, R_{n+1}| \cdots,$$

where the $D \times D$ matrices $A_{R_n}^{\sigma_n,\sigma'_n}$ are the same on every

site. We start at $\tau = 0$, for which $\tilde{\rho}(0) \propto \mathbb{1}$, and construct an MPO approximation for $\tilde{\rho}(\beta)$ by iteratively going through following 3 steps: (1) imaginary time-evolve with $H$ from $\tau$ to $\tau + \Delta\tau$, (2) adjust the normalization operator $N(\tau) \to N(\tau + \Delta\tau)$, and (3) truncate the virtual bond dimension of the MPO.

Let us now explain these 3 steps in more detail. The first step consists of a conventional TEBD update based on a Suzuki-Trotter approximation of $e^{-\Delta\tau H}$. As this step is standard (i.e. it is identical to a TEBD step for clean spin chains), we will not explain it in detail here. Let us only mention that the bond dimension of the MPO increases from $D$ to $DD_T$ during the TEBD update, with $D_T$ the bond dimension of the Trotter gates.

For the second step we trace out the physical spins to obtain $\mathrm{tr}_{\{\sigma\}}\left(N(\tau)e^{-(\tau+\Delta\tau)H}\right) = N_R(\tau)N_R^{-1}(\tau + \Delta\tau) =: \Lambda$, which is a diagonal MPO acting on the disorder qudits. The MPO $\Lambda$ is represented graphically in Fig. 1 (b). In practice, we find that the bond dimension of $\Lambda$, which is also $DD_T$, is highly redundant and can be truncated down to a much smaller value without any significant loss of precision. We now want to find an MPO representation of $\Lambda^{-1} = N_R^{-1}(\tau)N_R(\tau + \Delta\tau)$. We do this variationally by fixing a bond dimension $\chi$ of an ansatz MPO $\Lambda_M^{-1}$, which we take to be diagonal in the physical indices, and maximizing the fidelity between $\Lambda\Lambda_M^{-1}$ and the identity operator $\mathbb{1}_R$ [see Fig. 1 (c)]. This can be done directly in the thermodynamic limit by a straightforward generalization of the VOMPS algorithm (variational optimization of matrix product state) [4, 17, 18], which maximizes the fidelity per site. If the desired tolerance for minimizing the cost function is not achieved, we increase $\chi$. More details on the modified VOMPS algorithm can be found in the supplementary material [19]. At the end of step (2) we obtain $\tilde{\rho}(\tau + \Delta\tau) = N(\tau + \Delta\tau)e^{-(\tau+\Delta\tau)H} \approx (\Lambda_M^{-1} \otimes \mathbb{1}_\sigma)N(\tau)e^{-(\tau+\Delta\tau)H}$, which is an MPO of bond dimension $DD_T\chi$.

In the third and final step, we truncate this MPO back to an MPO of bond dimension $D$. Here we recall that the density matrix we actually wish to approximate is given in Eq. (1), which contains a sum over all disorder configurations. We therefore trace out the disorder qudits to obtain $\rho(\tau + \Delta\tau) = \mathrm{tr}_{\{R\}}\left([\prod_n P(R_n)]\,\tilde{\rho}(\tau + \Delta\tau)\right)$. Next we do a standard MPS-type truncation of $\rho(\tau + \Delta\tau)$ by going to the Schmidt basis for a bipartition of the system in a left- and right-infinite half, and keeping only the states corresponding to the $D$ largest Schmidt values [2]. The corresponding $D \times (DD_T\chi)$ isometry $W$ which implements this Schmidt truncation of $\rho(\tau + \Delta\tau)$ is then used to truncate the bond dimension of $\tilde{\rho}(\tau + \Delta\tau)$, i.e. the rank-$D$ projector $P_D = W^\dagger W$ is inserted on every virtual bond of $\tilde{\rho}(\tau + \Delta\tau)$. We finally redefine $\tau + \Delta\tau$ as $\tau$, and go back to step (1). This cycle is iterated until a particular imaginary time $\beta = T^{-1}$ is reached, after which we directly obtain an approximation for the density matrix in Eq. (1): $\rho \approx \mathrm{tr}_{\{R\}}\left([\prod_n P(R_n)]\,\tilde{\rho}(\beta)\right)$.

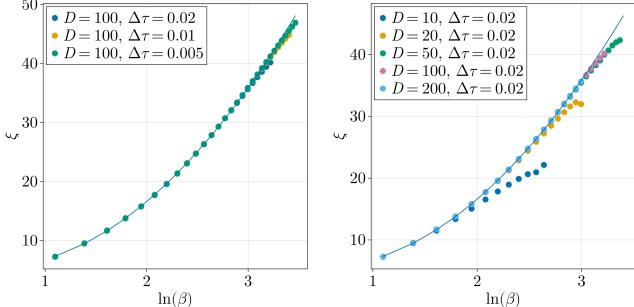
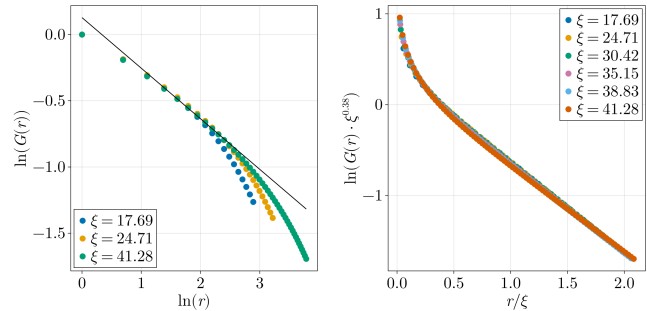

FIG. 2. Left: Correlation length of the average correlation function as a function of inverse temperature for different step sizes $\Delta\tau$. The blue line corresponds to a quadratic fit of the data for $\Delta\tau = 0.005$. Right: Correlation length as a function of inverse temperature for different bond dimensions. The blue line represents a quadratic fit of the data for $D = 200$. In both figures the correlation length was extracted from the transfer matrix eigenvalues.

FIG. 3. Left: The correlation functions for different correlation lengths with $D = 100$ and $\Delta\tau = 0.005$. The three data sets were obtained using $\beta$ equal to 8, 12 and 28. The black line represents the algebraic decay of the infinite-randomness fixed point. Right: Scaling collapse of the correlation functions for $D = 100$ and $\Delta\tau = 0.005$. In both figures we have extracted the correlation lengths from the scaling collapse.

Compared to the imaginary-time evolution of clean spin chains, the most important new step in our algorithm is to adjust the normalization $N(\tau)$ and the corresponding VOMPS step to find the MPO approximation of $\Lambda^{-1}$. We note that for sufficiently small $\Delta\tau$, $\Lambda = N_R^{-1}(\tau)N_R(\tau + \Delta\tau)$ is close to the identity, which has a trivial (MPO) inverse. Taking $\Delta\tau$ small enough, it should therefore always be possible to approximately invert $\Lambda$ with a low bond-dimension MPO. We also expect that after every imaginary-time step it is crucial to first restore the normalization before truncating the MPO. Otherwise, the truncation would not treat the different disorder sectors equally and could discard important contributions to $\rho$. In addition, the MPO truncation step itself is also different from the truncation step in the standard construction of the Gibbs state of $H$. The reason for this is that quite often one would like to work with many different disorder values and hence a large disorder qudit dimension, which would lead to a large bond dimension for the MPO representation of $e^{-\beta H}$. We therefore expect that by requiring the MPO to only approximate the density matrix with traced-out disorder qudits one can achieve a target numerical precision with much smaller bond dimension. We find that this expectation is indeed borne out in our numerical simulations presented below.

*Application to RTFIM* – We now illustrate our algorithm by applying it to the solvable [20–22] random transverse field Ising model (RTFIM). The Hamiltonian is given by

$$\mathcal{H}_{\mathrm{I}} = -\sum_n J_n \sigma_n^z \sigma_{n+1}^z + \sum_n h_n \sigma_n^x \,, \tag{5}$$

where $J_n$ and $h_n$ are uncorrelated random variables. For our simulations we fix $J_n = 1$ as our unit of energy and temperature, and take $h_n$ to be uniformly distributed between $[0.73, 1.3]$ with $N_D = 10$ disorder values. This distribution has

$$\delta = \frac{\ln\langle h_i \rangle - \ln\langle J_i \rangle}{\mathrm{Var}(\ln h_i) + \mathrm{Var}(\ln J_i)} \approx -0.049 \,, \tag{6}$$

which is close to the $\mathbb{Z}_2$-breaking critical point at $\delta_c = 0$ [21, 22], and is sufficiently broad to erase the clean Ising behavior already at short length scales [23]. We consider this model to be a highly non-trivial test for our algorithm, as the RTFIM has an *infinite-randomness* critical point where disorder-averaged quantities receive important contributions from rare regions. For the numerical simulations we set the tolerance for the inverse of $\Lambda$ at $10^{-8}$. Most data points required only a bond dimension of $\chi = 1$ or $\chi = 2$ for $\Lambda_M^{-1}$. Only at the lowest temperatures and largest step sizes $\Delta\tau$ we had to increase $\chi$ to at most 8.

From the MPO representation of $\rho$ we obtain the transfer matrix

$$T = \sum_R \sum_\sigma P(R) A_R^{\sigma,\sigma} \,, \tag{7}$$

whose spectrum determines the decay of disorder-averaged correlation functions. In particular, $\xi = [-\ln(\lambda_2/\lambda_1)]^{-1}$, with $\lambda_1$ and $\lambda_2$ respectively the largest and second largest eigenvalue of $T$, is the largest length scale encoded in the MPO. A characteristic property of the infinite-randomness fixed point is that the dynamical exponent is effectively infinite, leading to *activated dynamical scaling*: $\xi \sim [\ln \beta]^2$ [21]. In the left panel of Fig. 2 we show the correlation length obtained with $D = 100$ as a function of $\ln \beta$, for three different values of $\Delta\tau$. For $\Delta\tau = 0.005$ we see that $\xi$ almost perfectly follows the expected quadratic dependence on $\ln \beta$ up to $\beta \sim 30$, at which $\xi \sim 45$. For the larger imaginary-time increments $\Delta\tau = 0.01$ and $\Delta\tau = 0.02$ we see that $\xi$ deviates from the theoretical curve at smaller values of $\ln \beta$,

which we attribute to errors introduced by the Suzuki-Trotter decomposition of $e^{-\Delta\tau H}$. In the right panel of Fig. 2 we plot $\xi$ obtained with $\Delta\tau = 0.02$ and different bond dimensions as a function of $\ln\beta$. As expected, at smaller bond dimension $\xi$ deviates from the theoretical curve at smaller $\ln\beta$. We also see that the results are approximately converged for $D = 100$ (as the data obtained with $D = 100$ and $D = 200$ are indistinguishable), from which we conclude that finite bond dimension-errors have become negligible in this temperature range.

The left panel of Fig. 3 shows the disorder-averaged spin correlation function $G(r)$ (i.e. the $\sigma^z - \sigma^z$ correlation function) on a log-log plot for different values of $\xi$. These results were obtained at different temperatures using $\Delta\tau = 0.005$ and $D = 100$. We see that for larger $\xi$ increasingly more data points lie on the black straight line with a negative slope of $2 - \phi \approx 0.38$ (with $\phi$ the golden ratio), which is the exact exponent of the average spin correlation function at the infinite randomness fixed point [21, 22]. In the right panel of Fig. 3 we show a data collapse of the spin correlation function at different temperatures. This figure shows clear scaling: we find that $G(r)\xi^{2-\phi} = f(r/\xi)$ is a universal function of the ratio $r/\xi$, as predicted by theory [22]. Note that for both plots in Fig. 3 we have obtained $\xi$ via the data collapse with the exact exponent. This determines the correlation lengths at different $\beta$ up to an overall factor, which we have fixed by equating the smallest $\xi$ (obtained at $\beta = 8$) to the corresponding correlation length obtained from the transfer matrix spectrum. The two methods for obtaining the $\xi$'s (transfer matrix spectrum vs. scaling collapse of the correlation function) produce results that agree on the order of $\leq 10\%$ for the largest $\beta$. More numerical results are presented in the supplementary material [19], such as e.g. $\xi$ as a function of $\delta$, and $\xi(T)$ on the ordered side of the transition, which we confirm grows algebraically with varying exponent, i.e. $\xi \sim 1/T^{\alpha(\delta)}$ [22] (as opposed to $\xi \sim e^{\alpha'(\delta)/T}$ in the clean case). The supplementary material also contains further information on the performance of our algorithm.

*Conclusions* – Our work provides a proof-of-principle that the mixture of Gibbs states with different disorder configurations can be efficiently represented as a single translationally-invariant MPO. The specific algorithm developed here for constructing this MPO allows for many immediate improvements. For example, one could use the better-motivated density matrix truncation (DMT) method for the MPO representation of density matrices [24]. We also found that in the temperature range studied in this work the dominant errors seem to come from the Suzuki-Trotter decomposition. To mitigate these errors one could use the recently developed MPO representation of the cluster expansion of $e^{-\Delta\tau H}$ [25, 26], which produces errors that are systematically higher order in $\Delta\tau$. In principle, the cluster expansion method can also be used to reduce the number

of imaginary-time steps by increasing $\Delta\tau$ without significant loss of accuracy, but it is unclear whether the inversion of $\Lambda$, crucial for maintaining the correct normalization, can still be done efficiently in that case.

We have focused exclusively on disorder averages, but there seems to be no conceptual obstruction to generalize our method to calculate e.g. the second moment of the distribution of physical observables by representing the disorder average of two copies of the Gibbs state as an MPO. Whether this can still be done with numerically tractable bond dimensions is an interesting question for future work. An equally, perhaps even more, interesting question is whether there exists a variational version of our algorithm that can directly target disorder-averaged ground state properties. Finally, one drawback of our method is that we do not see a way to extract disorder-averaged von Neumann entropies [27–29] and typical correlations. But compared to average correlations the latter are more easily obtained via the standard method of explicitly sampling over disorder configurations.

*Acknowledgments.*— N.B. acknowledges stimulating discussions with Nicolas Laflorencie, Bram Vanhecke, Jeanne Colbois and Loic Herviou. This research was supported by the European Research Council under the European Union Horizon 2020 Research and Innovation Programme via Grant Agreement No. 101076597-SIESS (N.B.). W.T. is funded by the Research Foundation Flanders postdoctoral Fellowship 12AA225N.

———————

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

# Supplementary Material

## Extracting average properties of disordered spin chains with translationally invariant tensor networks

Kevin Vervoort[1], Wei Tang[1] and Nick Bultinck[1]

[1] *Department of Physics and Astronomy, Ghent University, Krijgslaan 281, 9000 Ghent, Belgium*

(Dated: May 1, 2025)

### Additional details of the VOMPS algorithm for MPO inversion

An essential part of the algorithm proposed in the main text is the inversion of the MPO $\Lambda$. To find the inverse of an MPO, we variationally optimize the fidelity between the product of the MPO and its candidate inverse and the identity operator. To perform this optimization, we make use of the tangent-space method [4, 17, 18]. More specifically, using the tangent-space method we derive a set of fixed-point equations, and the optimal solution for the MPO inverse can be obtained by iterating these fixed-point equations.

To start, we represent our ansatz for the inverse $\Lambda^{-1}$ as an MPO $\Lambda_M^{-1}$ of bond dimension $\chi$. In order to employ the tangent-space techniques, we reformulate the problem into an MPS problem by bending one of the vertical legs of the MPO downward, as shown in Eq. (8). This setup is very similar to that of the VOMPS algorithm [18], which approximates a product of an MPS and an MPO by another MPS with a smaller bond dimension. In our case, we look for an MPS such that the product of the MPS and an MPO approximates a given MPS.

$$\tag{8}$$

For later convenience, we will denote the MPS with filled circles and bent dashed line and the MPO with white circles and bent dashed line in Eq. (8) as

$$\tag{9}$$

The cost function for the variational optimization corresponds to the fidelity between the two states in Eq. (8), which is expressed as

$$\mathcal{L} = \ln\left( \frac{\langle\psi(\bar{A})|\Lambda^\dagger|\mathbb{1}\rangle\langle\mathbb{1}|\Lambda|\psi(A)\rangle}{\langle\psi(\bar{A})|\Lambda^\dagger\Lambda|\psi(A)\rangle} \right), \tag{10}$$

where we defined $|\mathbb{1}\rangle$ as the state representing the trivial MPS on the right hand side of Eq. (8), and $|\psi(A)\rangle$ is the MPS version of $\Lambda_M^{-1}$ (i.e. the MPO with filled circles and bent dashed line in Eq. (8) and Eq. (9)). The gradient of this cost function can be computed by taking the derivative with respect to $\bar{A}$. The optimal solution is then found as the point where the gradient vanishes. Writing out this gradient gives

$$\frac{\partial\mathcal{L}}{\partial\bar{A}} = \langle\partial_{\bar{A}}\psi(\bar{A})|\left( \Lambda^\dagger|\mathbb{1}\rangle - \frac{\langle\psi(\bar{A})|\Lambda^\dagger|\mathbb{1}\rangle}{\langle\psi(\bar{A})|\Lambda^\dagger\Lambda|\psi(A)\rangle}\Lambda^\dagger\Lambda|\psi(A)\rangle \right). \tag{11}$$

The optimality condition $\partial \mathcal{L}/\partial \bar{A} = 0$ can be reformulated as

$$\mathcal{P}_A \left( \Lambda^\dagger |\mathbb{1}\rangle - \frac{\langle \psi(\bar{A})|\Lambda^\dagger|\mathbb{1}\rangle}{\langle \psi(\bar{A})|\Lambda^\dagger \Lambda|\psi(A)\rangle} \Lambda^\dagger \Lambda |\psi(A)\rangle \right) = 0, \tag{12}$$

where the projector $\mathcal{P}_A$ is the projector that projects a state onto the tangent space of $|\psi(A)\rangle$, whose explicit form is given by

The condition (12) allows us to derive a set of consistency equation that the optimal solution should satisfy. Denoting the local tensor of the MPO $\Lambda$ as $O$, Eq.(12) leaves us with the following consistency equations for the optimal solution:

In these equations we have defined:

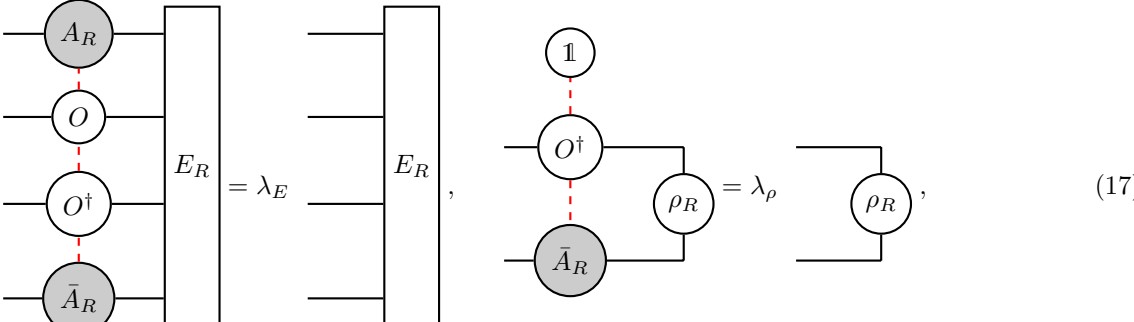

$$(17)$$

and similarly for $\rho_L$ and $E_L$.

In the following, we provide a detailed description of the algorithm.

(i) First we prepare an initial guess for $\Lambda_M^{-1}$. We then bring it to MPS form $|\psi(A)\rangle$ by bending one of the vertical legs of the MPO downward [c.f. Eq. (8)], and convert the MPS to the mixed canonical form. The virtual bond dimension for this MPS is the same as the inverse MPO, while the dimension of its open indices is the square of the original disorder qudit dimension.

(ii) We construct the environment tensors $\rho_L$, $\rho_R$, $E_L$, and $E_R$ by solving the eigenvalue equations (17).

(iii) By solving the linear equations in Eqs. (14) and (15), we can obtain a new set of $A_C$ and $C$.

(iv) From the new set of $A_C$ and $C$, we can obtain a new set of left- and right-canonical tensors $A_L$ and $A_R$, where we simply adopt the procedure used in the standard variational uniform MPS (VUMPS) algorithm [30].

(v) If $|\psi(A)\rangle$ is already precisely at the optimal point, then for the updated tensors $A_C$, $C$, $A_L$, and $A_R$, we expect both $\epsilon_L = \|A_C - A_L \cdot C\|$ and $\epsilon_R = \|A_C - C \cdot A_R\|$ to vanish. In practice, with our initial guess, this is rarely the case. Therefore, we need to return to step (ii), and iterate the process from step (ii) to step (iv) until both $\epsilon_L$ and $\epsilon_R$ go below a chosen tolerance.

(vi) Finally, after the iteration converges, we convert the optimized MPS $|\psi(A^\star)\rangle$ back into the MPO form, yielding the approximate inverse MPO $\Lambda_M^{-1}$.

### Performance of the algorithm for the MPO inversion

As argued in the main text, obtaining the inverse of $\Lambda$ as a low bond-dimension MPO should be possible as long as the imaginary-time steps are sufficiently small. We further check this expectation through the numerical results presented in Fig. 4. In Fig. 4(a), we observe only a moderate growth of the bond dimension of $\Lambda$ with $\beta$ (obtained for the RTFIM). Furthermore, this bond dimension can be made to increase more slowly by taking $\Delta\tau$ smaller. A small bond dimension for $\Lambda$ is consistent with this MPO being close to identity, suggesting that its inverse also admits a low bond-dimension MPO approximation.

In Fig. 4(b), we show the error introduced in the approximate inversion of $\Lambda$ for different inverse temperatures $\beta$. We see that the inverse becomes less accurate during the imaginary-time evolution. The accuracy of the MPO inversion can be improved by choosing a smaller $\Delta\tau$ or a larger $\chi$. In practice, we find that at small $\beta$ and $\Delta\tau$, an MPO with $\chi = 1$ can often represent the inverse with sufficient precision. In addition, we see that the inversion is more unstable for larger $\chi$, especially when the bond dimension of $\Lambda$ itself is small. However, at larger bond dimensions for $\Lambda$ the inversion becomes systematically better with increasing $\chi$.

Fig. 4(c) shows the correlation length of the average correlation function of the RTFIM obtained with different tolerances in the VOMPS algorithm used to approximately invert $\Lambda$. The data sets obtained with tolerances $\epsilon = 10^{-6}$ and $\epsilon = 10^{-8}$ show no noticeable differences, indicating that the error introduced by approximating the inverse of $\Lambda$ is negligible.

### Additional results for the Random Transverse Field Ising Model

The relevant quantity characterizing the behavior of the RTFIM is $\delta = [\ln\langle h_i \rangle - \ln\langle J_i \rangle]/[\text{Var}(\ln h_i) + \text{Var}(\ln J_i)]$. For $\delta > 0$, the model is in the paramagnetic phase, whereas the model exhibits long-range order and $\mathbb{Z}_2$ symmetry breaking for $\delta < 0$. On the ordered side of the quantum phase transition, the finite-temperature correlation length

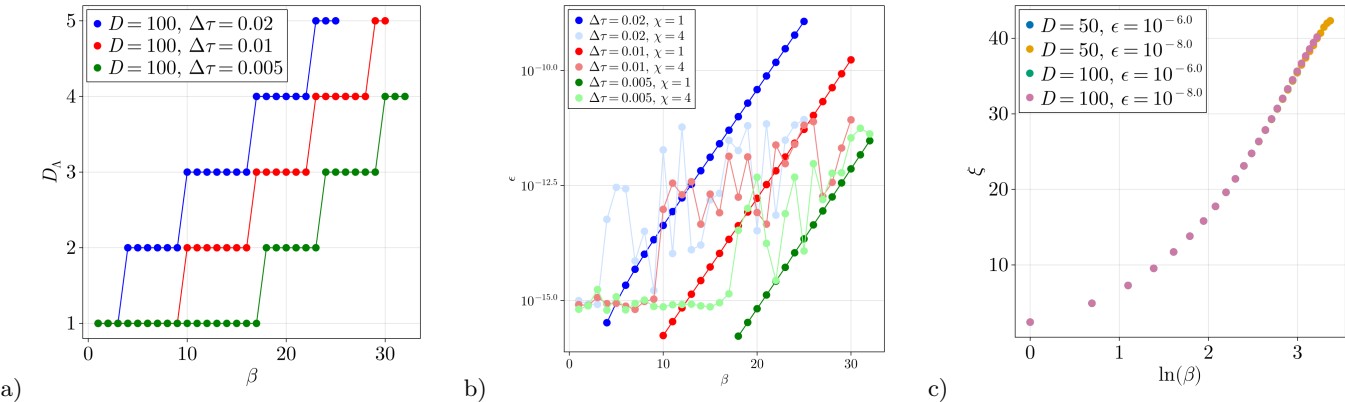

FIG. 4. (a) Bond dimension of $\Lambda$ for the RTFIM at inverse temperature $\beta$. (b) Error made in the inversion of $\Lambda$ for different bond dimensions $\chi$ of the MPO $\Lambda_M^{-1}$. The error is computed by summing all but the largest Schmidt values of $\Lambda\Lambda_M^{-1}$. (c) Correlation length of average correlation function in function of $\ln\beta$ for different tolerances for the VOMPS inversion step. The data was acquired with $\Delta\tau = 0.02$.

grows as $\xi \sim 1/T^{\alpha(\delta)}$ [22], i.e. the correlation length diverges as power of $1/T$ with a continuously varying exponent. This behavior is different from the clean Ising model, where the correlation length grows as $\xi \sim e^{\alpha'(\delta)/T}$. On the disordered side the correlation length is predicted to scale as $\xi \sim (\ln\beta)^2/[(\delta - \delta_c)^2(\ln\beta)^2 + a]$ [22], with $a$ some constant independent of $\delta$ and $\beta$. For large $\beta$ the correlation length saturates to $\xi \sim 1/(\delta - \delta_c)^2$. Motivated by these scaling considerations, we plot $\xi(\delta)/(A\ln(\beta/B))^2$ for different $\beta$ as a function of $\delta$ in Fig. 5. The constants $A$ and $B$ were determined from the quadratic fit to the correlation length data in the main text. The results were obtained with $J_n = 1$, and by taking $h_n$ to be uniformly distributed between $[0.7, h_{\max}]$ with $N_D = 10$ disorder values. To change $\delta$ we performed simulations with different values of $h_{\max}$ ranging from 1.28 to 1.5. The imaginary time step size used for all simulations was $\Delta\tau = 0.02$.

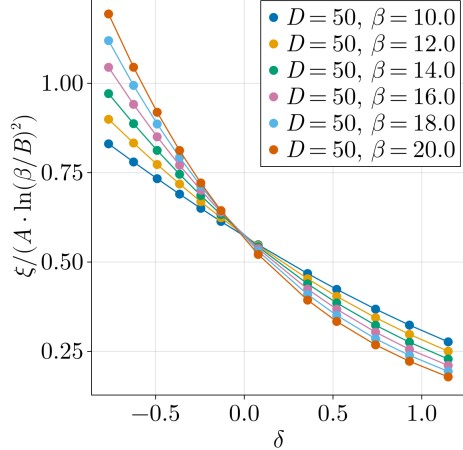

FIG. 5. Correlation length of the disorder-averaged correlation function as a function of $\delta$ for different temperatures. The different curves cross around $\delta \approx 0$, which corresponds to $h_{\max} \approx 1.34$.

Fig. 6 shows in more detail the growth of the correlation length with $\ln\beta$ in both the ordered phase (left panel) and disordered phase (right panel). The results confirm the expected scaling behavior of the correlation length. In the left panel of Fig. 6 we see that the correlation length on the ordered side grows as a power of $\beta$. Furthermore, the slope of the curves on the log-log plot changes with $\delta$, confirming the continuously varying nature of the exponent. In the right panel of Fig. 6 the correlation length on the disordered side of the transition is shown. The data shows the expected initial growth $\propto (\ln\beta)^2$ for small $\ln\beta$. Furthermore, the correlation length at large $\beta$ increases with decreasing $\delta$ as expected.

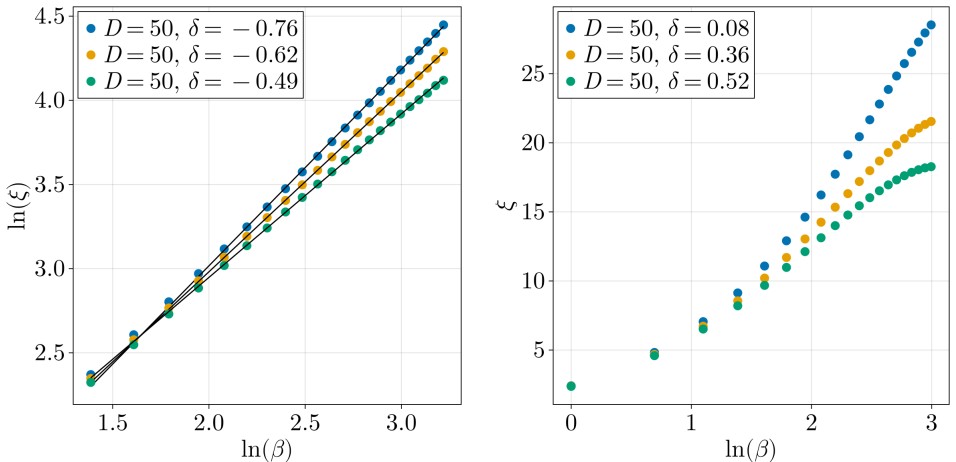

FIG. 6. Left: Correlation length in function of inverse temperature for different $\delta$ on the ordered side of the quantum phase transition. The black lines represent linear fits to the data. Right: Correlation length in function of inverse temperature for different $\delta$ on the disordered side.