# Peer review of "Extracting average properties of disordered spin chains with translationally invariant tensor networks"

_SciPost Physics_

## Round 1 · Referee Report · Natalia Chepiga (Referee 1) · 2025-9-16

Strengths

Interesting idea addressing a timely problem

Weaknesses

see the report

Report

The authors develop a tensor-network approach to simulate quantum systems with disorder avoiding the need to run simulations multiple times to sample over various disorder configurations. The authors achieve this by introducing an auxiliary disorder qudits. They benchmark the algorithm on the paradigmatic model of disordered quantum systems – random transverse field Ising model.

I think the problem is timely and this manuscript is an important first step toward the solution. However, there are several reservations that prevent me to recommend the manuscript in its current form to be published in SciPost Physics:

  1. First of all, this is a methodology paper – it does not report new physics, but the algorithm that potentially can solve problems in future more efficiently. This is fine, but for the methodology paper, I would expect more thorough analysis of complexity and computational costs of the new method compare to the existing ones. I would naively say that the authors save a lot of computing run time by the price of higher memory costs and sacrificing the access to some information (typical scaling, probability distribution etc). This has to be properly discussed, ideally the gain and loss compared to the standard techniques have to be quantified. I would also recommend to move the discussion on the performance of the algorithm from supplementary materials to the main text.

  2. My second concern is the usage of the random Ising model as a benchmark. The authors made a very big assumption when going from the continuous distribution of random coupling/field to a discrete with a very few values. This assumption has to be justified. Ising model is very dangerous choice for benchmarking – everything works on Ising model; it is exactly solvable. I am not at all convinced that discrete and small number of random parameters would work in general. And this is the main bottleneck of the method – it would be extremely expensive to go to a larger values of the discrete parameter or to make them different on every site (simple with open boundary conditions, but seems non-trivial for transitionally invariant infinite tensor network).

  3. The authors analyzed only the correlation function and comment that extraction of the entanglement entropy is complicated by construction of the algorithm. But what about the energy gap and/or the gap between the Schmidt values?

  4. The authors claim that they lose access to the typical values and over statistical distribution. But I wonder whether this information is in fact contained in the final network that the authors optimize. I wonder if there is a way to post-process the final optimized MPS to extract the distribution and how computationally costly this could be?

  5. The authors use known critical exponents of the infinite randomness critical point to produce a data collapse. But what would be the strategy and performance of the method if these exponents are not known or non-universal (as in the case of random XXZ chain)?

In addition, I have a few minor comments that the authors might want to consider:

  1. In the intro, the authors write that “randomness can cause entanglement to be inhomogeneously distributed… which significantly increases the number of sweeps needed for convergence”. In my experience the DMRG convergence for disordered system is in fact much faster than for the clean one. Could the authors provide a references reporting a slow convergence (or the authors had in mind the works of Ref.7-9)?

  2. At the end of the first column on page 1 the authors refer to a “relatively small bond dimension”. It would be useful to provide at least the order of magnitude here.

  3. The authors claim that they use a modified version of TEBD to ensure the correct normalization. More details here (on the normalization conditions and how TEBD helps with it) would be useful

  4. Fig.3 appears on top of p.3, but it is first discussed only on p.4

10 a. The authors say they use “uniformly distributed” random field h_n “between [0.73, 1.3]”. Is it in the linear or in the log scale?

10 b. Also, how do the authors “fine-tune” the system to the critical point? The value of delta in Eq.6 seems very large: in numerical simulations it is typical to keep this value below 10^{-3}-10^{-4}.

  1. The comment on “important contributions from rare regions” requires a reference.

Recommendation

Accept in alternative Journal (see Report)

---

## Round 1 · Referee Report · Anonymous (Referee 2) · 2025-9-22

Disclosure of Generative AI use

The referee discloses that the following generative AI tools have been used in the preparation of this report:

"DeepL write", to correct the English.

Strengths

1- The paper proposes a novel approach to simulate one-dimensional disordered systems with translation-invariant MPOs 2- The test case is solvable and yet highly non-trivial 3- The presentation is transparent

Weaknesses

1- Minor improvements could be made to clarify the presentation and context (see report)

Report

[Summary:] This work presents an original approach to compute disorder-average properties of random-spin chains using translation-invariant tensor networks, by exploiting the restoration of translation invariance upon disorder averaging. Although the idea of using ancilla qdits to represent the disorder and perform the average was already present in the literature, this was so far mainly used for dynamics. The present work highlights how to get finite temperature disorder average results, focusing on two ways of obtaining the correlation lengths. The manuscript is clear and concise and tests the approach on a non-trivial test case, where key signatures of the presence of the infinite-randomness fixed point are recovered. Except for the minor comments below, it seems to me that the work easily meets the expectations and criteria for this journal.

[Comments/questions:] I have minor comments and questions. The first two, though minor, should be addressed. The others are written only out of curiosity and interest in the work, or are marginal suggestions for presentation improvements. They may be disregarded if the Authors do not find them helpful or relevant.

1- In the text, (\Lambda = \mathrm{Tr}_{\sigma} (N(\tau)e^{-(\tau + \Delta \tau)H})). However, Fig. 1 and its caption suggest that (\Lambda) is obtained by tracing out the spin degrees of freedom in (\tilde{\rho}(\tau)). It seems to me that this distinction is very small, but important. Could the Authors check this and either correct the figure accordingly or clarify why this distinction does not matter?

2- Context/outlook: It seems to me that the specific context could be slightly better spelled out for the reader, with a small consequence for the outlook. (a) The authors cite the work of Paredes et al, but it would be nice to highlight how the present work is different in spirit. Similarly, I think the work https://scipost.org/SciPostPhys.6.3.031 may be given as part of the context. (b) In the outlook, the authors ask the question of whether a variational version of their algorithm could directly target ground-state properties. With proper context, it is clear that the question is about the properties of their algorithm rather than about targeting ground states, which was already discussed in the previous references. (c) Slightly less directly connected, it may be interesting to comment on the existence of numerical SDRG (see e.g. https://arxiv.org/pdf/2501.02643 for a recent example on a related model) for ground states, and the fact that they can access notably dynamical exponents and typical behavior. This would be an opportunity to highlight better what the present method brings. I also think it may make sense to either cite the review by Igloi & Monthus or the finite-temperature work of Young (PRB 1997) on the transverse-field Ising chain.

3- (Optional) Entanglement spectrum: It seems natural to perform the Schmidt decomposition on (\rho) rather than (\tilde{\rho}). However, this density matrix is a prior of a different type than that which occurs in translation-invariant problems. It is therefore tempting to ask if : (a) the entanglement spectrum decays fast enough to justify truncation; (b) there is any interpretation of the validity of the approach (e.g., validity of "typicality") / an expectation of where it should fail (e.g., Bose glass in another model? ). Otherwise stated, could it be that the approach works better close to the IRFP than deep in the localized phases ?

4 - (Optional) Possible marginal improvements to the presentation (in order of appearance in the text):

a) Although it is well-known in the community, a specific reference for the statement "The community has reached a point at which low-temperature equilibrium properties of most local spin and fermion Hamiltonians on a one-dimensional lattice can be studied with rather limited computational resources, including many gapless systems" may be helpful to newcomers.

b) (\tilde{\rho}) is introduced in Eq. (3), but its connection to (\rho) is only explicitly clarified at the bottom of page 2. Clarifying this connection from the start may help the reader.

c) Fig. 3, right panel: The figure is slightly difficult to read/interpret (in particular because of course, it is a good collapse). I have two suggestions : (i) could the authors also indicate the inverse temperature? (ii) Could the lowest temperature be shown with crosses or other symbols that do not hide the larger temperatures?

Requested changes

1- Check if a modification is needed with respect to (\Lambda) in Figure 1 (see point 1 in the report). 2- Better clarify what the existing literature has already established (see point 2 in the report).

Recommendation

Ask for minor revision

---

## Editorial Decision

in_refereeing